# Comparative analysis of a rapid diagnostic test and scoring tools for ESBL detection in *Enterobacterales* bloodstream infections for optimizing antimicrobial therapy

Sam R. Andrews,[1] Tristan T. Timbrook,[2,3] Mark A. Fisher,[4,5] Brandon J. Tritle[1]

**ABSTRACT** Extended-spectrum beta-lactamase (ESBL)-producing *Enterobacterales* bloodstream infections (BSI) are associated with increased cost, morbidity, and mortality compared with BSI by non-ESBL-producing bacteria. Phenotypic susceptibility data from blood cultures can take up to 72 hours to result, leading to delays in appropriate antimicrobial administration and unnecessary carbapenem usage. We sought to evaluate the performance of a rapid diagnostic test (RDT) and two published diagnostic scoring tools for predicting ceftriaxone resistance in BSI by *Enterobacterales*. This was a retrospective observational cohort study evaluating adult patients with BSI by select *Enterobacterales* species (*Escherichia coli*, *Klebsiella oxytoca*, *Klebsiella pneumoniae* group, *Proteus mirabilis*, and *Salmonella* spp.). We compared the performance of a multiplex-PCR panel by BioFire (BCID2) with two diagnostic scoring tools for predicting ceftriaxone resistance. Among 356 patients, ceftriaxone resistance was observed in 41 of 356 (11.5%) isolates. BCID2 accurately predicted ceftriaxone susceptibility in 99.2% of isolates (92.7% sensitivity and 100% specificity), compared with <90% for the two scoring tools (*P* < 0.05 for both). In ceftriaxone-resistant isolates (*n* = 41), BCID2 led to fewer instances of undertreatment with ceftriaxone compared with scoring tools (7.3% vs 68.3% and 92.7%). In ceftriaxone-susceptible isolates (*n* = 315), BCID2 led to no instances of overtreatment with carbapenems compared with 3.8% and 5.7% with the scoring tools. The BCID2 panel outperformed both ESBL scoring tools in our population. Local evaluation of RDT and scoring tools should be performed in conjunction with institutional implementation.

**IMPORTANCE** Our study addresses a significant issue in the medical and scientific community—the delayed administration of appropriate antimicrobial treatments due to the time-consuming process of phenotypic susceptibility data collection in gram-negative bloodstream infections. Our research indicates that a multiplex PCR rapid diagnostic test (RDT) significantly outperformed two clinical scoring tools in predicting ceftriaxone susceptibility. Multiplex PCR also led to reduced instances of undertreatment with ceftriaxone and minimized overtreatment with carbapenems. Furthermore, multiplex PCR demonstrated high sensitivity and specificity in predicting ceftriaxone susceptibility. The results of our study underscore the potential RDTs to reduce the time to appropriate antimicrobial therapy, leading to improved patient outcomes and reduced healthcare costs.

**KEYWORDS** ESBL, rapid diagnostics, risk prediction, risk score, bacteremia

Address correspondence to Sam R. Andrews, sam.andrews@ucsf.edu.

Tristan T. Timbrook is an employee and stockholder of BioMérieux. No other authors have any financial disclosures.

An estimated 2 million bloodstream infections (BSI) occur in North America and Europe annually and are associated with 250,000 deaths (1). With antimicrobial resistance on the rise, extended-spectrum beta-lactamase-producing *Enterobacterales*

(ESBL-E) have been identified by the CDC and WHO as organisms of concern (2, 3). From 2013 to 2019, cases of ESBL-E infections were estimated to have increased by 50%. Infections of ESBL-E are associated with increased cost, morbidity, and mortality when compared with non-ESBL-producing bacteria; they are also associated with delayed administration of appropriate antimicrobial therapy (3–5). Susceptibility data from blood cultures can take up to 72 hours to result, contributing to these delays and, consequently, poorer clinical outcomes (6). The risk of undertreatment of resistant organisms must be continually weighed against the repercussions of overtreatment with carbapenems, the treatment of choice for invasive infections by ESBL-E (7).

Several scoring tools utilizing risk factors for ESBL-E BSI have been developed to aid clinicians predict their presence and determine the need for empiric treatment with carbapenems. Augustine and colleagues (8) created a weighted scoring system utilizing three independent risk factors for ESBL-E BSI: (i) recent outpatient gastrointestinal/genitourinary procedure, (ii) recent course of beta-lactams or fluoroquinolone, and (iii) prior infection/colonization with ESBL-E. They concluded that patients with scores ≥3 (Approach 1) or critically ill patients with a score of 1–2 (Approach 2) should receive carbapenem therapy (8). Additionally, Lee and colleagues (9) developed a scoring tool that identified nursing home residence, recent antimicrobial therapy, recent invasive procedures, and frequent emergency department (ED) visits as risk factors for ESBL-E, and recommended empiric carbapenem therapy with a score ≥2. Both of these methods were found to have high negative predictive values (NPV, 97%–99%), although a limitation of these studies is that they were conducted in populations with a relatively low incidence of ESBL-E. Furthermore, these scoring tools had poor positive predictive values (PPV, 33%–40%), which could lead to overtreatment with carbapenem therapy (8, 9).

In April 2021, the University of Utah Health began utilizing the BCID2 platform for rapid identification of organisms in blood cultures. This new RDT detects the most common gene conferring ESBL resistance, $bla_{CTX-M}$ (CTX-M). Despite this, providers frequently select meropenem out of concern for other mechanisms of ceftriaxone resistance. The use of a similar RDT by Verigene at the Detroit Medical Center and the University of Maryland Medical Center was found to be effective in predicting ceftriaxone susceptibility of select *Enterobacterales* (10). The purpose of this study was to validate the usage of the BCID2 platform in predicting ceftriaxone susceptibility at the University of Utah Health and compare its discrimination performance against two ESBL diagnostic scoring tools.

## MATERIALS AND METHODS

This was a retrospective observational study evaluating adult patients with BSI by select *Enterobacterales* species across the University of Utah Health system. Patients were included if they had at least one positive blood culture with *Escherichia coli*, *Klebsiella oxytoca*, *Klebsiella pneumoniae*, *Proteus* spp., or *Salmonella* spp., which are all species of *Enterobacterales* detected by the BCID2 panel. While BCID2 can also detect *Enterobacter cloacae*, *Klebsiella aerogenes*, and *Serratia marcescens*, patients with BSI by these organisms were excluded due to the possibility of AmpC beta-lactamase production. Patients were excluded if carbapenemases were detected. Patients who had more than one species of *Enterobacterales* identified were excluded. However, detection or isolation of non-*Enterobacterales* organisms (i.e., *Staphylococcus* spp., *Bacteroides*, etc.) did not preclude patients from this study. If a patient had multiple episodes of BSI by *Enterobacterales*, only the first episode within the study period was included. Pregnant and incarcerated patients were excluded.

This study closely followed a methodology previously described (10). Briefly, we assessed the validity of three methods for determining whether a ceftriaxone or a carbapenem would have been indicated in gram-negative BSI: (i) the CTX-M result from the BCID2 panel, (ii) the ESBL prediction score by Augustine and colleagues (8), and (iii)

the ESBL prediction score by Lee and colleagues (9). Time 0 was defined as the time blood cultures were drawn, at which point all patients were hypothetically started on empiric ceftriaxone. Blood cultures were presumed positive at 24 hours, and patients were managed hypothetically by each method. A patient was switched to a carbapenem if (i) CTX-M was detected on the BioFire BCID2 panel (suggesting ESBL phenotypic resistance prior to finalized susceptibilities), (ii) the ESBL prediction score by Augustine and colleagues (8) was ≥3, or (iii) the ESBL prediction score by Lee and colleagues (9) was ≥2. Otherwise, the patient was continued on ceftriaxone. Definitive antibiotic therapy was determined at 72 hours when phenotypic antibiotic susceptibility reports were available.

For each diagnostic method, the sensitivity, specificity, PPV, and NPV of each method were determined for predicting ceftriaxone susceptibility. Additionally, the theoretical number of patients who unnecessarily escalated to a carbapenem or undertreated with ceftriaxone was calculated. A theoretical number of carbapenem days were determined. As an exploratory outcome, ertapenem susceptibility was determined for ESBL isolates. Covariates collected included previously identified risk factors (see Table 1) (8, 9). Accuracy of the diagnostic approaches was compared using chi-square tests. Performance was compared using receiver operating characteristics (ROC) curve analysis. All statistical analyses were performed in R (R Foundation for Statistical Computing, Vienna, Austria).

**TABLE 1** Patient characteristics and risk factors[a]

| Characteristics | Ceftriaxone susceptible (n = 315) | Ceftriaxone resistant (n = 41) | Overall (N = 356) |
|---|---|---|---|
| Age, years (IQR) | 63 (47,73) | 57 (44,65) | 62 (47,72) |
| Female, n (%) | 173 (54.9) | 22 (53.7) | 195 (54.8) |
| Location, n (%) | | | |
| ED | 200 (63.5) | 19 (46.3) | 219 (61.5) |
| Floor | 61 (19.4) | 17 (41.5) | 80 (22.4) |
| ICU | 32 (10.2) | 4 (9.8) | 36 (10.1) |
| Ambulatory | 19 (6.0) | 1 (2.4) | 21 (5.9) |
| Organism, n (%) | | | |
| *Escherichia coli* | 223 (70.8) | 32 (78.0) | 255 (71.6) |
| *Klebsiella pneumoniae* group | 55 (17.5) | 9 (22.0) | 64 (18.0) |
| *Klebsiella oxytoca* | 15 (4.8) | 0 | 15 (4.2) |
| *Proteus mirabilis* | 14 (4.4) | 0 | 14 (3.9) |
| *Salmonella* spp. | 8 (2.5) | 0 | 8 (2.2) |
| Monomicrobial bacteremia, n (%) | 285 (90.5) | 38 (92.7) | 323 (90.7) |
| Ceftriaxone resistance, n (%) | n/a | n/a | 41 (11.5) |
| Risk factors, n (%) | | | |
| Admission from nursing home | 12 (3.8) | 4 (9.8) | 16 (4.5) |
| Hospitalization in prior 4 weeks | 79 (25.1) | 13 (31.7) | 92 (25.8) |
| Diabetes mellitus history | 124 (39.4) | 17 (41.5) | 141 (39.6) |
| Recent GI/GU procedure within 30 days | 35 (11.1) | 8 (19.5) | 43 (12.1) |
| Invasive procedure in previous 4 weeks | 17 (5.4) | 1 (2.4) | 18 (5.1) |
| No. of prior BL/FQ courses in previous 90 days | | | |
| 0 | 216 (68.6) | 17 (41.5) | 233 (65.4) |
| 1 | 71 (22.5) | 15 (36.6) | 86 (24.2) |
| ≥2 | 28 (8.9) | 9 (22.0) | 37 (10.4) |
| Antibiotic exposure in previous 4 weeks | 84 (26.7) | 19 (46.3) | 103 (40.2) |
| ESBL infection or colonization in prior year | 5 (1.6) | 10 (24.4) | 15 (4.2) |
| ≥3 ED visits within the previous year | 35 (11.1) | 5 (12.2) | 40 (11.2) |

[a]BL, beta-lactam; ED; emergency department; ESBL, extended-spectrum beta-lactamase; FQ, fluoroquinolone; GI, gastrointestinal; GU, genitourinary; ICU, intensive care unit; IQR, interquartile range.

## RESULTS

Between April 2021 and January 2023, 356 patients were included. The majority of blood cultures were drawn in the emergency department (219, 65.1%). The most frequently isolated organism was *Escherichia coli* (255, 71.6%), followed by *Klebsiella pneumoniae* group (64, 18.0%), *Klebsiella oxytoca* (15, 4.2%), *Proteus mirabilis* (14, 3.9%), and *Salmonella* species (8, 2.2%). Phenotypic ceftriaxone resistance was observed in 41 (11.5%) patients, which included 32 (78%) *Escherichia coli* and 9 (22.0%) *Klebsiella pneumoniae* group isolates. All ceftriaxone-resistant isolates were susceptible to ertapenem. Characteristics and risk factors are summarized in Table 1. Augustine and Lee score distributions are displayed in Table 2.

BCID2 correctly predicted ceftriaxone susceptibility in 99.2% of patients, which was significantly higher than both the Augustine and Lee scoring tools (88.8% and 88.4%, respectively, $P < 0.05$, for both; Table 3). Areas under the ROCs are shown in Fig. 1. CTX-M was detected in 38 (10.7%) patients, leading to escalation to a carbapenem at 24 hours. By the Augustine and Lee scoring tools, escalation of therapy occurred in 25 (7%) and 21 patients (5.9%), respectively. Failure to escalate for ceftriaxone-resistant isolates ($n = 41$) occurred in three patients (7.3%) utilizing BCID2, which was fewer than by the Augustine (28, 68.3%) and Lee (38, 92.7%) scoring tools (Table 4). The number of carbapenem days (normalized to 1,000 patient-days) was 136, 135, and 136 for BCID2, Augustine, and Lee scoring tools, respectively.

## DISCUSSION

BCID2 performed well at our institution with high sensitivity, specificity, PPV, and NPV. It outperformed both the Augustine and Lee scoring tools with our local evaluation, which had unacceptably low PPVs and sensitivities. The Lee scoring tool performed particularly poorly, with a sensitivity of only 7.3% and PPV of 14.3%, compared with a previously published sensitivity of 84.6% and PPV of 40.4% (9). These dissimilar findings might be explained by differences in patient populations; ESBL prevalence in our population was higher (11.5%) when compared with the populations in the Augustine and Lee scoring tool studies (4.6% and 5.7%, respectively). Further, Lee and colleagues (9) only included cases of community-onset bacteremia, while this study included patients at all stages of care. Lastly, we did not calibrate the risk scores to reflect our population's risk factors and incidence of ESBL (11).

Varying performance of diagnostic scoring tools has been previously described in the literature. In one systematic review examining scoring tools for predicting ESBL presence and carbapenem resistance in *Enterobacterales* bloodstream infections, subsequent external validation studies reported a median sensitivity and specificity of 32% and 90% for ESBL prediction, which were lower than that of their original studies (55% and 94%, respectively) (12). In another study at several community hospitals, the implementation of an ESBL diagnostic scoring tool in *Enterobacterales* bloodstream infections led to a significant decrease in mean time to appropriate antibiotic administration (78 hours before to 46 hours after implementation) (13). Altogether, these findings underscore the potential clinical relevance of these tools, particularly for NPV among low incidence settings, and the necessity of local validation of diagnostic scoring tools to ensure effective implementation.

Overall carbapenem usage was similar among the three methods, which can likely be explained by the large proportions of patients being undertreated and overtreated

**TABLE 2** ESBL scores[a]

| ESBL score ($N = 356$) | Augustine, no. (%) | Lee, no. (%) |
|---|---|---|
| 0 | 209 (58.7) | 220 (61.8) |
| 1 | 94 (26.4) | 115 (32.3) |
| 2 | 28 (7.9) | 19 (5.3) |
| ≥3 | 25 (7.0) | 2 (0.6) |

[a]ESBL, extended-spectrum beta-lactamase.

**TABLE 3** Accuracy of diagnostic methods[a]

| N = 356 | Cutoff | Appropriately predicted CTX susceptibility | Sensitivity | Specificity | PPV | NPV |
|---|---|---|---|---|---|---|
| BCID2 | CTX-M + | 99.2% | 92.7% | 100% | 100% | 99.1% |
| Augustine | 3 | 88.8% | 31.7% | 96.2% | 52.0% | 91.5% |
| Lee | 2 | 84.3% | 7.3% | 94.3% | 14.3% | 88.7% |

[a]CTX, ceftriaxone; NPV, negative predictive value; PPV, positive predictive value.

by the Augustine and Lee scoring tools. Up to 68.3% and 92.7% of ceftriaxone-resistant isolates (n = 41) were undertreated with ceftriaxone, and 3.8% and 5.7% of ceftriaxone-susceptible isolates (n = 315) were overtreated with carbapenems by the Augustine and Lee scoring tools, respectively.

A strength of this study was that it included patients across the entire health system, validating the use of BCID2 in all patients with BSI by select *Enterobacterales* at the University of Utah Health. However, given that rates of ESBL-E at institutions in different regions are likely to differ, internal validation studies of rapid diagnostics platforms should be conducted to ensure accuracy for CTX-M targets. Further, prevalence of ESBL genes other than CTX-M, which are not detected by the BCID2 panel, vary by region, highlighting the need to assess local prevalence before implementing treatment guidelines based on the CTX-M target. One limitation of this study is that ESBL-E was defined by the presence of ceftriaxone resistance, but other mechanisms such as plasmid-mediated AmpC beta-lactamases can confer resistance to third generation cephalosporins in these organisms. This was not examined as testing for the presence of AmpC beta-lactamases is not routinely performed in most clinical microbiology labs.

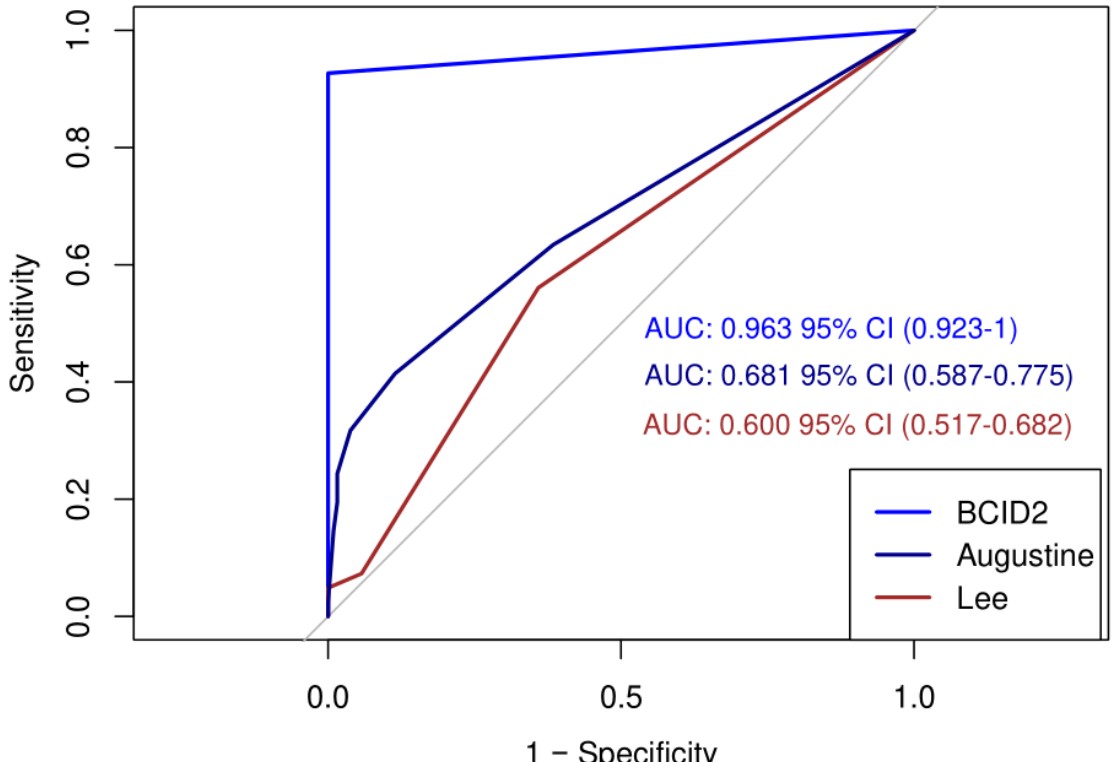

**FIG 1** Areas under the ROC curves by diagnostic method. Graphical illustration of test sensitivity vs specificity for BCID2 and Augustine and Lee diagnostic tools. Abbreviations: AUC, area under the curve; ROC, receiver operating characteristic.

**TABLE 4** Theoretical treatment outcomes[a]

| N = 356 | Overtreatment (n = 315) | Undertreatment (n = 41) |
|---|---|---|
| BCID2 | 0 (0) | 3 (7.3) |
| Augustine | 12 (3.8) | 28 (68.3) |
| Lee | 18 (5.7) | 38 (92.7) |

[a]Overtreatment = patient escalated to an empiric carbapenem at 24 hours (susceptibility = ceftriaxone susceptible); undertreatment = patient continued on ceftriaxone at 24 hours (susceptibility = ceftriaxone resistant).

A limitation of this study is that we did not investigate the clinical impact of implementing the BCID2 rapid diagnostics platform. Further studies are needed to elucidate the clinical impact. Additionally, generalizability of results is unclear given the varying epidemiology of CTX-M ESBL production detected by the BCID2 panel in addition to other ESBL types (i.e., TEM and SHV-type ESBLs). However, our results were similar to those reported previously with other rapid diagnostic technologies with similar targets (10), and CTX-M has been estimated to account for almost 80% of ESBL-E in the United States and Europe (14, 15).

## Conclusions

While both the Augustine and Lee ESBL scoring tools performed poorly in our patient population, BCID2 was found to be a highly sensitive and specific diagnostic tool with a high PPV and NPV for predicting ceftriaxone susceptibility at our institution. Local performance evaluations are important for supporting the use of genotypic antimicrobial resistance rapid diagnostic tests and risk scoring.

## ACKNOWLEDGMENTS

This study was performed without financial support.

S.R.A., T.T.T., M.A.F., and B.J.T. all participated in the concept and design of the study. Acquisition, analysis, and interpretation of the data, along with drafting of the manuscript, were performed by S.R.A. Critical revision of the manuscript was performed by S.R.A., T.T.T., M.A.F., and B.J.T.

## AUTHOR AFFILIATIONS

[1]Department of Pharmacy, University of Utah Health, Salt Lake City, Utah, USA
[2]BioMérieux, Salt Lake City, Utah, USA
[3]College of Pharmacy, University of Utah Health, Salt Lake City, Utah, USA
[4]Department of Pathology, University of Utah Health, Salt Lake City, Utah, USA
[5]Associated Regional and University Pathologists Laboratories, University of Utah Health, Salt Lake City, Utah, USA

## PRESENT ADDRESS

Sam R. Andrews, UCSF Health, San Francisco, California, USA

## AUTHOR ORCIDs

Sam R. Andrews  http://orcid.org/0009-0001-8303-9909
Tristan T. Timbrook  http://orcid.org/0000-0002-4542-6203

## AUTHOR CONTRIBUTIONS

Sam R. Andrews, Conceptualization, Data curation, Formal analysis, Investigation, Methodology, Project administration, Resources, Software, Validation, Writing – original draft, Writing – review and editing | Tristan T. Timbrook, Conceptualization, Formal analysis, Investigation, Methodology, Supervision, Writing – review and editing | Mark

A. Fisher, Conceptualization, Data curation, Investigation, Methodology, Resources, Supervision, Writing – review and editing.

## ADDITIONAL FILES

The following material is available online.

Open Peer Review

**PEER REVIEW HISTORY (review-history.pdf).** An accounting of the reviewer comments and feedback.

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
