## [Reviewer comments · Microbiology Spectrum]

Microbiology Spectrum

Comparative Analysis of a Rapid Diagnostic Test and Scoring Tools for ESBL Detection in Enterobacterales Bloodstream Infections for Optimizing Antimicrobial Therapy

Sam Andrews, Tristan Timbrook, Mark Fisher, and Brandon Tritle

Corresponding Author(s): Sam Andrews, University of California San Francisco

Review Timeline:

Submission Date:	September 10, 2023
Editorial Decision:	October 16, 2023
Revision Received:	November 3, 2023
Accepted:	November 10, 2023

Editor: Krisztina Papp-Wallace

Reviewer(s): The reviewers have opted to remain anonymous.

Transaction Report:

DOI: <https://doi.org/10.1128/spectrum.03131-23>

October 16, 2023

Dr. Sam Rose Andrews
University of California San Francisco
505 Parnassus Ave
San Francisco, CA 94143

Re: Spectrum03131-23 (Optimizing Antimicrobial Therapy in Enterobacterales Bloodstream Infections: Comparative Analysis of a Rapid Diagnostic Test and Scoring Tools for ESBL Detection)

Dear Dr. Sam Rose Andrews:

Link Not Available

Sincerely,

Krisztina Papp-Wallace

Journals Department
Reviewer comments:

Reviewer #1 (Comments for the Author):

Thank you for conducting this study comparing the BCID2 panel with risk scoring tools. I feel like there are a few different concepts being brought into this study that make the overall message diluted a bit. Some of the concepts I'm picking up include stewardship impact on RDT use/effectiveness, ensuring local validation of publicly available risk scoring tools, and accuracy of RDTs to scoring tools. While these concepts are all linked, I would highly encourage the authors to focus on 1 or maybe 2 of those concepts at most to strengthen the take-home message. That said, the take-home message of this paper seems to be that BCID2 is accurate and should be used (which is known information), hence, spinning the paper a touch more to RDT accuracy vs just using a scoring tool (with or without validating it) might make this paper a bit more novel/useful to existing literature.

Reviewer #2 (Comments for the Author):

In this study, the authors evaluated the performance of a RDT (BCID2) in predicting ceftriaxone resistance in BSI caused by select ESBL-E. The authors also compared the performance of BCID2 in predicting ceftriaxone resistance compared to two diagnostic scoring tools. The manuscript is well-written and easy to follow. Below are some possible points of clarification/elaboration specific to the current manuscript:

1. The scoring tools by Augustine and Lee were published in 2017 and it is unclear the clinical relevance of either scoring tool, especially since the current paper's methods and results align closely with the paper by Cwengros et al (2020; DOI: 10.1093/ofid/ofaa278) that identified similar benefits with another commonly used RDT (Verigene) compared to the same scoring tools. Consider adding some background information in the introduction or the discussion sections that address the current real-world relevance of these scoring tools. For example, have they been referenced in other recent ESBL-E papers that would lead clinicians to use the scoring tools over the seemingly more common practice of RDT results and CTX-M identification?
2. Please provide a breakdown of information specific to the PPV and NPV differences between groups. Was there a commonality between the RDT and scoring tools that consistently led to either correct identification or misidentification among all three (e.g., organisms [ESBL-E], source of infection, etc)? From those results, is there a better scoring tool that could be created using data from the current study and that from Cwengros et al. that would facilitate appropriate therapy in the first 24 hours of therapy?
3. Please clarify how ceftriaxone susceptibility/resistance was defined prior to the release of MIC data. Was it strictly based on CTX-M presence or absence?
4. Authors may consider incorporating a two-tier NPV and PPV approach. Tier 1 would be CTX-M results and Tier 2 would be CRO-R vs. CRO-S.
5. Please update the text in lines 74-75 to fix the presumed typo.

Staff Comments:

Preparing Revision Guidelines

Please return the manuscript within 60 days; if you cannot complete the modification within this time period, please contact me. If you do not wish to modify the manuscript and prefer to submit it to another journal, please notify me of your decision immediately so that the manuscript may be formally withdrawn from consideration by Microbiology Spectrum.

In this study, the authors evaluated the performance of a RDT (BCID2) in predicting ceftriaxone resistance in BSI caused by select ESBL-E. The authors also compared the performance of BCID2 in predicting ceftriaxone resistance compared to two diagnostic scoring tools. The manuscript is well written and easy to follow. Below are some possible points of clarification/elaboration specific to the current manuscript:

1. The scoring tools by Augustine and Lee were published in 2017 and it is unclear the clinical relevance of either scoring tool, especially since the current paper's methods and results align closely with the paper by Cwengros et al (2020; DOI: 10.1093/ofid/ofaa278) that identified similar benefits with another commonly used RDT (Verigene) compared to the same scoring tools. Consider adding some background information in the introduction or the discussion sections that addresses the current real-world relevance of these scoring tools. For example, have they been referenced in other recent ESBL-E papers that would lead clinicians to use the scoring tools over the seemingly more common practice of RDT results and CTX-M identification?
2. Please provide a breakdown of information specific to the PPV and NPV differences between groups. Was there a commonality between the RDT and scoring tools that consistently led to either correct identification or misidentification among all three (e.g., organisms [ESBL-E], source of infection, etc)? From those results, is there a better scoring tool that could be created using data from the current study and that from Cwengros et al. that would facilitate appropriate therapy in the first 24 hours of therapy?
3. Please clarify how ceftriaxone susceptibility/resistance was defined prior to release of MIC data. Was it strictly based on CTX-M presence or absence?
4. Authors may consider incorporating a two-tier NPV and PPV approach. Tier 1 would be CTX-M results and Tier 2 would be CRO-R vs. CRO-S.
5. Please update text in lines 74-75 to fix the presumed typo.

Below, we have listed each comment and corresponding response, including selections of manuscript text where the relevant changes have been made. Reviewer comments are denoted by boldface text and direct responses to the reviewers are denoted by italics. All edits to the manuscript have been highlighted in yellow.

Reviewer Comments:

Reviewer #1 (Comments for the Author):

Thank you for conducting this study comparing the BCID2 panel with risk scoring tools. I feel like there are a few different concepts being brought into this study that make the overall message diluted a bit. Some of the concepts I'm picking up include stewardship impact on RDT use/effectiveness, ensuring local validation of publicly available risk scoring tools, and accuracy of RDTs to scoring tools. While these concepts are all linked, I would highly encourage the authors to focus on 1 or maybe 2 of those concepts at most to strengthen the take-home message. That said, the take-home message of this paper seems to be that BCID2 is accurate and should be used (which is known information), hence, spinning the paper a touch more to RDT accuracy vs just using a scoring tool (with or without validating it) might make this paper a bit more novel/useful to existing literature.

Thank you for your thoughtful feedback on our manuscript. We appreciate your comments and the effort you've invested in reviewing our work.

You correctly pointed out that there are multiple concepts embedded within our study. We acknowledge that intertwining these concepts might make the primary message less clear to our readers. We intended to provide a comprehensive analysis but understand the importance of maintaining clarity and focus.

We have adjusted the positioning in the abstract, and conclusion to downplay the stewardship impact. We believe both the performance of RDT compared to culture and risk scoring tools are critical messages to maintain and thus have left those as is.

Reviewer #2 (Comments for the Author):

In this study, the authors evaluated the performance of a RDT (BCID2) in predicting ceftriaxone resistance in BSI caused by select ESBL-E. The authors also compared the performance of BCID2 in predicting ceftriaxone resistance compared to two diagnostic scoring tools. The manuscript is well-written and easy to follow. Below are some possible points of clarification/elaboration specific to the current manuscript:

We appreciate your review and considerate feedback.

1. The scoring tools by Augustine and Lee were published in 2017 and it is unclear the clinical relevance of either scoring tool, especially since the current paper's methods and results align closely with the paper by Cwengros et al (2020; DOI: 10.1093/ofid/ofaa278) that identified similar benefits with another commonly used RDT (Verigene) compared to the same scoring tools. Consider adding some background information in the introduction

or the discussion sections that address the current real-world relevance of these scoring tools. For example, have they been referenced in other recent ESBL-E papers that would lead clinicians to use the scoring tools over the seemingly more common practice of RDT results and CTX-M identification?

We have added more background information in the discussion on the clinical relevance of the scoring tools with improved context more clearly delineated with the reference of a recent systematic review and meta-analysis on this topic (<https://pubmed.ncbi.nlm.nih.gov/37760748/>) which reflects the commonality of the development and real-world use of the risk score tools.

2. Please provide a breakdown of information specific to the PPV and NPV differences between groups. Was there a commonality between the RDT and scoring tools that consistently led to either correct identification or misidentification among all three (e.g., organisms [ESBL-E], source of infection, etc)? From those results, is there a better scoring tool that could be created using data from the current study and that from Cwengros et al. that would facilitate appropriate therapy in the first 24 hours of therapy?

Derivation of a novel risk scoring or prediction model was beyond the scope of this paper. As noted above, we have added expanded background on the use of these tools in the discussion with a referenced systematic review. Additionally, to this point we have added discussion around the challenges in performance of risk prediction tools, particularly with external validation given most risk factors are only low to moderately sensitive or specific beyond previous resistant organism infection or colonization.

3. Please clarify how ceftriaxone susceptibility/resistance was defined prior to the release of MIC data. Was it strictly based on CTX-M presence or absence?

Yes, the ceftriaxone susceptibility prior to release of MIC data was based on CTX-M presence or absence. We have added a sentence to the methods on this clarification.

4. Authors may consider incorporating a two-tier NPV and PPV approach. Tier 1 would be CTX-M results and Tier 2 would be CRO-R vs. CRO-S.

While this is an interesting approach to analyzing our data, we feel that this is beyond the scope of our research, as this places the focus on the diagnostic scoring tools rather than comparing the accuracy of a rapid diagnostic test with that of the scoring tools.

5. Please update the text in lines 74-75 to fix the presumed typo.

Fixed.

Re: Spectrum03131-23R1 (Comparative Analysis of a Rapid Diagnostic Test and Scoring Tools for ESBL Detection in Enterobacteriales Bloodstream Infections for Optimizing Antimicrobial Therapy)

Dear Dr. Sam Rose Andrews:

Your manuscript has been accepted, and I am forwarding it to the ASM production staff for publication. Your paper will first be checked to make sure all elements meet the technical requirements. ASM staff will contact you if anything needs to be revised before copyediting and production can begin. Otherwise, you will be notified when your proofs are ready to be viewed.

Sincerely,
Krisztina Papp-Wallace
Editor
Microbiology Spectrum